electrical engineering/mechanical engineering

aluminium conductor steel reinforced,
formed aluminium conductor steel reinforced,
mechanical characteristics, finite-element
modelling

**Author for correspondence:**
Bo Yan
e-mail: boyan@cqu.edu.cn

# Study on mechanical characteristics of conductors with three-dimensional finite-element models

Jiaqiong Liu[1], Bo Yan[1], Guizao Huang[1], Zheyue Mou[1], Xin Lv[2] and Haibing Zhang[3]

[1]College of Aerospace Engineering, and [2]College of Civil Engineering, Chongqing University, Chongqing 400044, People's Republic of China
[3]State Grid Electric Power Research Institute of Chongqing Electric Power Company, Chongqing 401123, People's Republic of China

JL, 0000-0002-6709-3876; BY, 0000-0002-1251-2674

Refined three-dimensional (3D) finite-element (FE) models of typical aluminium conductor steel reinforced (ACSR) and formed aluminium conductor steel reinforced (FACSR) with structural details to simulate their static and dynamic characteristics are proposed. Taking into account the elastoplastic behaviour of the aluminium wires, the tensile mechanical properties and coupling between tension and torsion of the two types of conductors under tensile loading are numerically investigated. Furthermore, dynamic responses of two transmission lines, in which the refined 3D segment models and equivalent beam models of the two types of conductors are used, after ice-shedding are numerically simulated and the dynamic characteristics of the conductors are analysed. Finally, based on the numerical simulation results, the fatigue lives of the aluminium wires are estimated and the wear between the wires is discussed. It is revealed that taking into account the structural details of the conductors in the strength design of transmission lines is necessary, and the mechanical characteristics of FACSR are better than those of the ACSR in both static and dynamic situations.

## 1. Introduction

Transmission conductors are key components in the power grid and their safety is very important for the normal operation of the power supply system. The conductors of aluminium conductor steel reinforced (ACSR) are widely used, and a new type of conductors known as formed aluminium conductor steel reinforced (FACSR) are obtaining more and more application in China. The stranded

structures of both types of conductors give rise to coupling between tensional and torsional deformations even when only tensile load is applied along their axial direction. Moreover, more complicated coupling of tensile, bending and torsional deformations may be induced when the transmission conductor lines are subjected to dynamic loads such as wind and ice-shedding. Fatigue of the wires and fretting wear between the internal wires under dynamic loads may reduce lifetime of the conductors. It is necessary to study the mechanical characteristics of the two types of conductors under static and dynamic loads.

There are some theoretical studies on mechanical properties of wire ropes, cables, conductors and other helical structural strands. Knapp [1] derived an element stiffness matrix for helical structural cables subjected to tension and torsion based on the principle of energy conservation, but only linear elastic property of the wires is considered. Lanteigne [2] theoretically studied the tension, torsion and bending of ACSR under static loading and indicated that bending deformation is independent of axial stretching and torsion deformation, while axial stretching is coupled with torsion deformation. Costello [3] investigated the quasi-static behaviour of a wire rope under uniaxial stretching, bending and torsion, and the contact stress between the strands during the deformations. However, the simplifications of these models may lead to obvious deviation of the results from the real behaviour of the strand structures.

Utting & Jones [4,5] experimentally studied the quasi-static axial tensile behaviour of a typical $1 + 6$ strand, which was then extensively investigated by other authors. Akhtar & Lanteigne [6] performed tension–torsion measurements on 10 types of multi-strand conductors, including ACSR, AACSR (aluminium alloy conductor steel reinforced) and ACAR (aluminium conductor alloy reinforced), and obtained the tensile yield strengths and ultimate strengths of these conductors, but no elastoplastic data of the individual wires are given. McConnell & Zemke [7] tested the static bending stiffness of multi-strand conductors and indicated that the bending stiffness increases with axial load. Papailiou [8,9] measured the dynamic tensile and bending properties of some conductors and set up a simulation model, but the stress and deformation of the wires in the conductors were not studied.

The finite-element (FE) method has been used to study the mechanical behaviour of conductors by some authors. Jiang *et al.* [10] created a concise FE model for wire ropes based on the geometric helical symmetry of the ropes, but the boundary conditions of the model are difficult to apply. Judge *et al.* [11] developed three-dimensional (3D) elastoplastic FE models of stranded steel cables subjected to quasi-static axial loading and discussed the mechanical characteristics of the cables under working stress and the breaking load. Some researches [12–14] have developed 3D FE models to analyse the stress distribution of an optical ground wire subjected to a prescribed displacement. Frigerio *et al.* [15] studied the axial force-elongation behaviour of an AAAC (all aluminium alloy conductor) with a 3D FE model, in which the residual stress resulting from manufacture process is included. Foti & Roseto [16] proposed a formulation to model the elastoplastic properties of steel strands subjected to axial-torsional loads, and the obtained results are consistent with experimental ones. Yu *et al.* [17] examined the contact and lateral load behaviour of a seven-wire steel strand by means of the FE method. In addition, the beam elements were used by some authors [18,19] to investigate the mechanical behaviour of stranded cables, but the refined local stress in the wires cannot be revealed.

By means of the FE model discretized with beam elements, Lalonde *et al.* [20,21] analysed the strain and stress amplitude at critical regions of a conductor-clamp system under cycle bending loads, studied the dynamic bending characteristics and estimated the fatigue lives of the conductors. Qi & McClure [14,22] analysed the stress of a stranded conductor with a 3D nonlinear FE model of an ACSR conductor-clamp system under fretting fatigue conditions, and developed a multi-axial fatigue lifing methodology to estimate local fretting fatigue strength of the electrical conductors. However, the 3D model was not used in an integral model to reveal the responses when the whole span overhead line is subjected to external loads. There are some numerical and experimental works on tension variation in conductors during vibration, such as dynamic responses of transmission lines after ice-shedding [23–25], dynamic swing of transmission lines under steady and stochastic wind [26] and galloping of iced transmission lines [27]. However, in these researches the conductors are equivalent to uniform cables without considering the effects of the stranded details of the conductors on their stress, strength, fatigue and wear.

In this paper, refined elastoplastic FE models of two types of conductors, ACSR and FACSR, with structural details discretized with 3D solid elements are proposed, and the efficiency of the models is verified by comparing the results with those obtained by other authors. Using the numerical models, the elastoplastic deformation and stress distributions of the two types of conductors under static loads are investigated. Furthermore, using the mixed models, in which the transmission conductors are discretized with refined 3D models and equivalent beam models, the dynamic responses of these

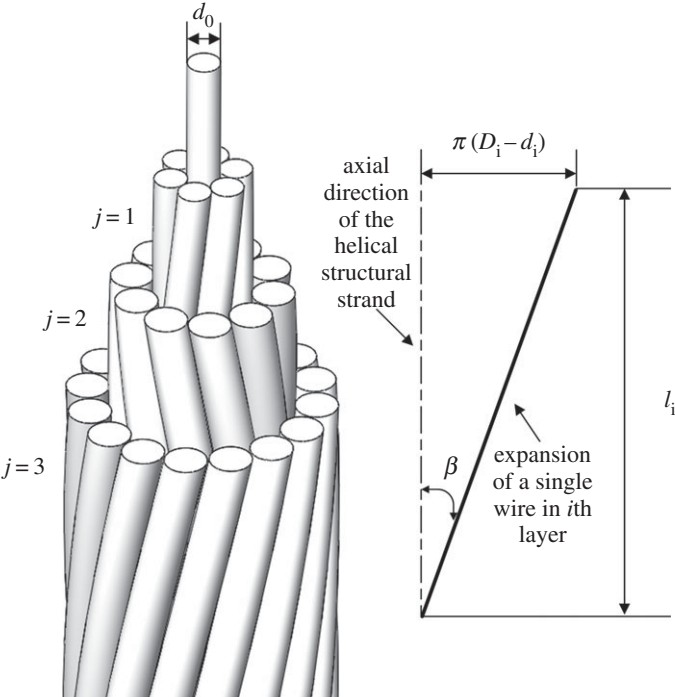

**Figure 1.** Helical structural strand and the plan of a wire in $i$th layer.

transmission conductors after ice-shedding are in turn analysed, based on which the fatigue lives of the aluminium wires are estimated and wear between the wires of the two conductors are discussed. The obtained results may provide instructions and foundations for the strength design and type selection of conductors in transmission lines.

# 2. FE modelling of helical structural strands

## 2.1. Geometrical characteristics of helical structural strands

As shown in figure 1, a helical structural strand is composed of several helical layers that are laid helically over a straight central core [28]. The central core is always round while the helical layers are round, trapezoidal or with other shapes, and the adjacent layers are laid in opposite directions. The external diameter $D_i$ of $i$th layer of a strand with $k$ layers can be determined by

$$D_i = d_0 + 2\sum_{j=1}^{i} d_j \quad (i \le k),$$

$$(2.1)$$

where $d_0$ is the diameter of the central core, $d_j$ is the diameter of a wire in $j$th layer. The lay angle $\beta_i$ of a wire in $i$th layer is given by

$$\tan \beta_i = \frac{\pi(D_i - d_i)}{l_i},$$

$$(2.2)$$

where $l_i$ is the pitch length of a wire in $i$th layer, $d_i$ is the diameter of a wire in $i$th layer, and the lay ratio of a wire in $i$th layer $m_i$ is defined by

$$m_i = \frac{l_i}{D_i}.$$

$$(2.3)$$

The guideline is a cylindrical helix curve and the contour line is a circle or trapezoid, so a layer of wires can easily be created by scanning in a 3D modelling software.

**Table 1.** Geometrical and mechanical parameters of a seven-wire strand.

| layer | wire number | wire diameter (mm) | lay angle (°) | Young's modulus (MPa) | Poisson's ratio | Yield stress (MPa) | Tangent modulus (MPa) |
|---|---|---|---|---|---|---|---|
| core wire | 1 | 3.94 | — | 188 000 | 0.3 | 1540 | 24 600 |
| helical wire | 6 | 3.73 | 11.8 | 188 000 | 0.3 | 1540 | 24 600 |

## 2.2. FE modelling of helical structural strands

To numerically simulate the mechanical behaviour of a conductor line with structural details, 3D geometrical model of a segment of the conductor is firstly created by means of the Solidworks software, and is then imported into the ABAQUS/CAE to establish the FE model.

General contacts between the wires are defined in the FE model. The general contact algorithm in ABAQUS/Standard offers the capabilities to model surface-to-surface contact, edge-to-surface contact and edge-to-edge contact, which can capture all the existing contacts in the model. The 'hard contact' of pressure-overclosure model and isotropic Coulomb friction are adopted. The model is discretized with eight-node linear brick C3D8R elements, with reduced integration and hourglass control. This type of element is suitable for contact analysis, especially when elastoplastic deformation is taken into account in the analysis.

## 2.3. Verification of FE modelling

A typical helical seven-wire strand was experimentally studied by Utting & Jones [4,5] and numerically simulated by Jiang et al. [10] with a so-called concise FE model, in which only a fraction of a strand slice was analysed. A segment of this seven-wire strand with length of 200 mm under axial tension is used to verify the FE modelling method presented in this paper. The geometrical and mechanical parameters of the strand are listed in table 1, and the FE model discretized with 65 700 C3D8R elements is shown in figure 2a.

A linear hardening model expressed with yield stress and tangent modulus is employed to model the elastoplastic behaviour of the wires, and the friction coefficient between the wires is set to be 0.115 [11]. All degrees of freedom at one end of the strand are fixed while the other end is coupled with a reference point, to which an axial load is applied. Due to the helical symmetry of the strand, the segment under tension will rotate even though only axial load is applied. To analyse the torsional behaviour of the segment, two cases are simulated. In one case, the rotation of the coupled point around the axis is fixed during loading and the end surface of the segment uniformly deforms in axial direction, which is denoted as fixed-end case. In the other case, all degrees of freedom of the coupled point are free and the end can rotate around its axis during loading, which is denoted as free-end case.

The deformation and stress of the FE model are then analysed by means of the ABAQUS software and the obtained relations between the axial load and the axial strain are compared with the experimental and numerical results by Utting & Jones [4,5] and Jiang et al. [10] as shown in figure 2b. It is seen that the numerical results by the FE model presented in this paper are consistent with those by the experiment and the concise FE model. Compared with the concise FE model, the boundary conditions of the model presented in this paper can be applied easily for the simulation.

In addition, Yu et al. [17] numerically studied bending behaviour of a segment of seven-wire strand with an initial axial pre-tension under lateral loading. The helical wire diameter is 3.72 mm, the lay angle is 17.03°, and the other geometric and mechanical parameters of the model are the same with those in table 1. The total length of the segment is 180 mm. One end of the segment is fixed. On the other end, an axial pre-tension load is firstly applied and lateral displacement is then gradually applied. Two cases are numerically simulated, one with pre-tension of 400 MPa and lateral displacement of 12.5 mm, and the other with pre-tension of 600 MPa and lateral displacement of 25 mm. The pre-tension was applied by means of dynamic relaxation method (in LS-DYNA) in the work of Yu et al. [17], while it is applied directly on the end in this paper.

The 3D FE model of the segment of the seven-wire strand is created and its deformation and stress distributions are then analysed by means of the ABAQUS/Standard. The FE model and bending deformation of the segment under pre-tension 600 MPa and lateral displacement 25 mm are shown in

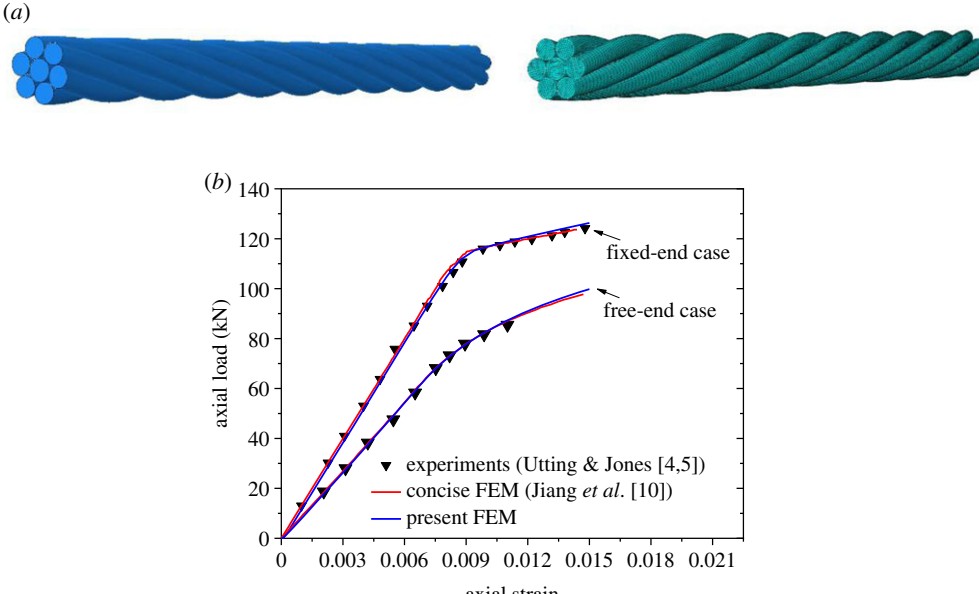

**Figure 2.** FE modelling of seven-wire strand under axial tension. (*a*) FE model; (*b*) relations between axial load and axial strain.

figure 3*a*, and the relationships between the lateral load and lateral displacement at the right end obtained by the method presented in this paper and Yu *et al.* [17] is shown in figure 3*b*. It is seen that under the same load the results obtained in this paper are close to those of Yu *et al.* [17], and the largest relative error is less than 4.0%.

Based on the two numerical examples, it is concluded that the FE modelling of the helical structural strand created in this paper can be employed to investigate the mechanical behaviour of conductors with the details of helical structure.

# 3. Static characteristics of conductors under axial tension

## 3.1. Tensile tests of steel and aluminium wires

The conductors JL/G1A-300/40 and JLX/G1A-300/40, which are typical conductors of ACSR and FACSR, respectively, are selected to investigate the mechanical characteristics of the two types of conductors. The mechanical properties of the steel and aluminium wires are firstly measured by simple tension tests.

The JL/G1A-300/40 conductor has one round steel core, one layer of round steel wires and two layers of round aluminium wires, and the JLX/G1A-300/40 conductor consists of one round steel core, one layer of round steel wires and two layers of trapezoidal aluminium wires. The structural parameters of these two types of conductors are shown as table 2. It is noted that the total cross-section areas of the two conductors are equal and the average lay ratio of each layer specified in reference [29] is employed into the models.

The wire samples with length of 300 mm were used for the simple tension tests, and three aluminium wires in each layer and seven steel wires were taken from the real conductors. The tests were carried out by a SANS electronic universal testing machine as shown in figure 4*a*, and the typical stress–strain curves of the aluminium and steel wires are shown in figure 4*b*. Based on the curves, Young's modulus, yield stress and ultimate stress of the aluminium and the steel wires were determined. The mechanical parameters of the wires determined by the average values of the samples of the two conductors are listed in table 3.

## 3.2. FE modelling of conductors

To investigate the static mechanical characteristics of the two types of conductors, the deformation and stress distributions of the two segments of conductors under axial tension are numerically simulated by means of the ABAQUS software. The length of both conductor models is 143.5 mm, and the length verification shows that this length is appropriate. One end of a segment is fixed, and uniform tension is applied to the other end, which is coupled with a point, and the load is applied to this point. The

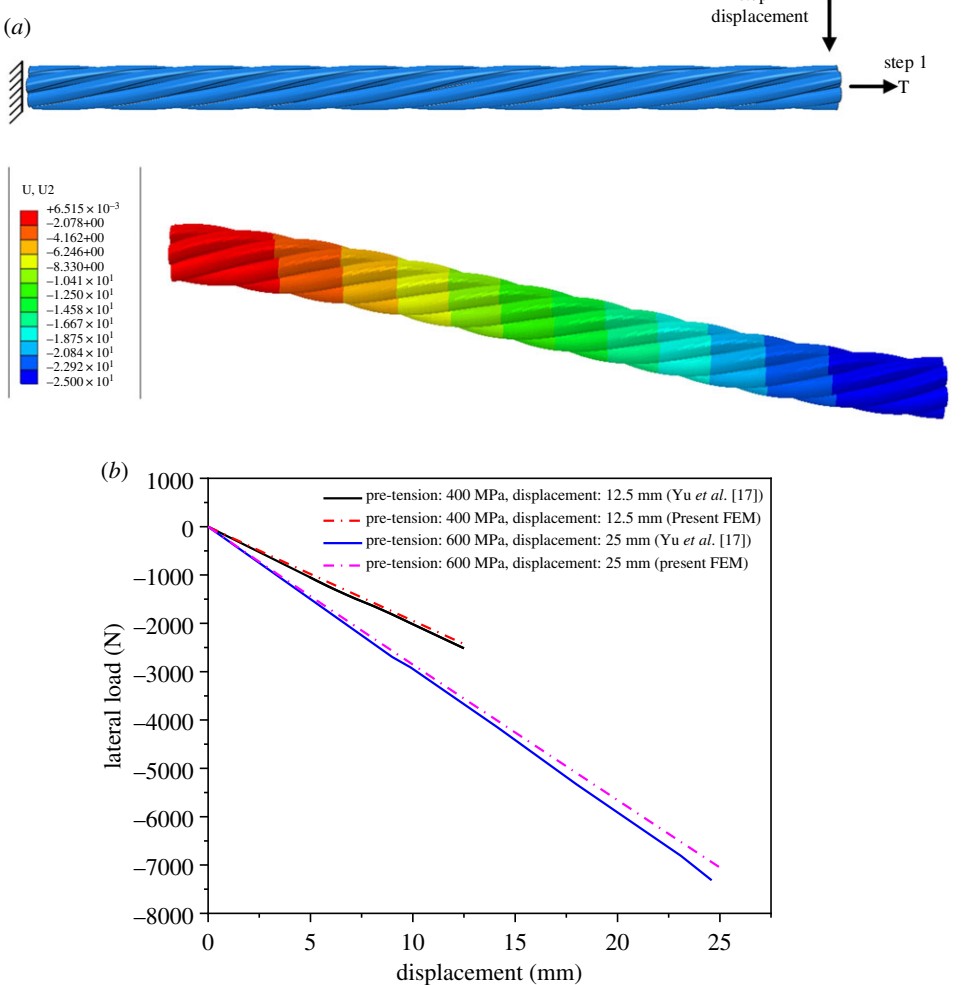

**Figure 3.** Model and deformation of seven-wire strand numerically determined. (a) Model of seven-wire strand; (b) relations between lateral load and displacement of seven-wire strand under pre-tension and bending load.

**Table 2.** Structural parameters of JL/G1A-300/40 and JLX/G1A-300/40. Note: C1: JL/G1A-300/40; C2: JLX/G1A-300/40.

| layer number | wire number | | wire diameter (mm) | | cross-section area (mm²) | | lay ratio | |
|---|---|---|---|---|---|---|---|---|
| | C1 | C2 | C1 | C2 | C1 | C2 | C1 | C2 |
| core | 1 | 1 | 2.66 | 2.66 | 38.9 | 38.9 | — | — |
| layer 1 | 6 | 6 | 2.66 | 2.66 | | | 21 | 21 |
| layer 2 | 9 | 8 | 3.99 | — | 300.9 | 301.0 | 13 | 13 |
| layer 3 | 15 | 10 | 3.99 | — | | | 12 | 12 |

fixed-end case and free-end case mentioned in §2.3 are discussed. The axial loads are gradually applied and the total loads for the two conductors are 60 kN.

The geometrical models of the two conductor segments are created by means of Solidworks as shown in figure 5a, and the models are then imported into ABAQUS/CAE to establish the FE models shown in figure 5b. The JL/G1A-300/40 model is discretized with 818 600 C3D8R elements and JLX/G1A-300/40 model with 544 300 elements, and mesh convergence arrives for these two models. General contacts between the wires are defined in the FE models. The friction coefficients for aluminium–aluminium, steel–aluminium and steel–steel contacts are, respectively, set to be 0.5, 0.5 and 0.3 [30]. The real

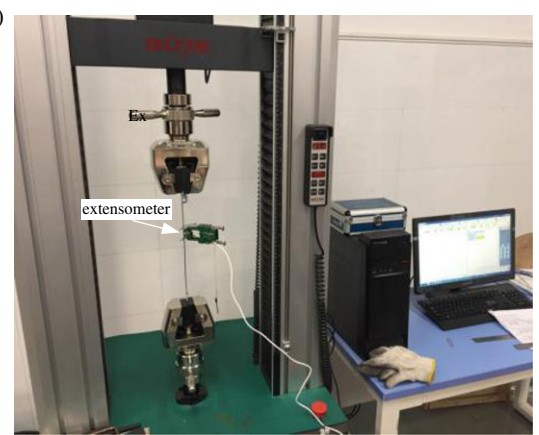

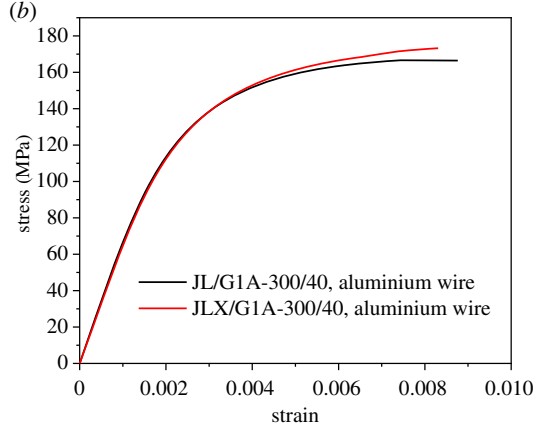
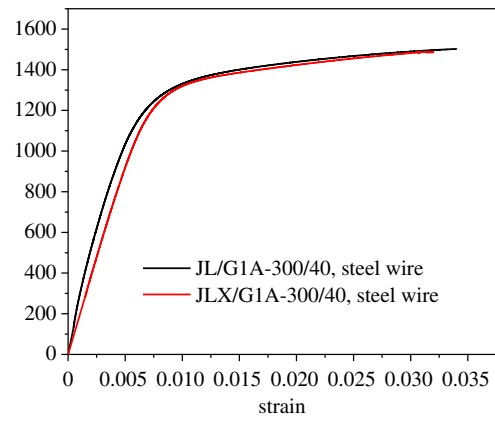

**Figure 4.** Tension tests of conductors JL/G1A-300/40 and JLX/G1A-300/40. (*a*) Test system; (*b*) stress–strain curves of typical individual aluminium and steel wires.

**Table 3.** Mechanical parameters of individual aluminium and steel wires.

| conductor | wire | Young's modulus (MPa) | yield stress (MPa) | ultimate stress (MPa) |
|---|---|---|---|---|
| JL/G1A-300/40 | aluminium | 69 028 | 148.6 | 166 |
| | steel | 209 258 | 1172 | 1489 |
| JLX/G1A-300/40 | aluminium | 69 028 | 151.3 | 173 |
| | steel | 209 258 | 1172 | 1489 |

stress–strain curves of the aluminium wires obtained by experiments are imported into the ABAQUS software to express the elastoplastic relation of the material, and elastic model of the steel wires is used in the FE analysis.

## 3.3. Deformation and stress characteristics of conductors

### 3.3.1. Overall tensile behaviour of conductors

Due to the helical symmetry of the conductors, twist will take place though only tensile load is applied to one end of a segment. To investigate the overall characteristics of the two types of conductors under axial tensile load, the torque at the end of the conductor, the twist angle of the conductor and the equivalent axial strain are extracted from the FE results. The relations of the equivalent axial strain with the axial load, the torque and the twist rate determined by the numerical simulations and Costello's theory [3] are shown as figure 6. It is noted that the materials are assumed as linear elasticity without considering plastic deformation in Costello's theory.

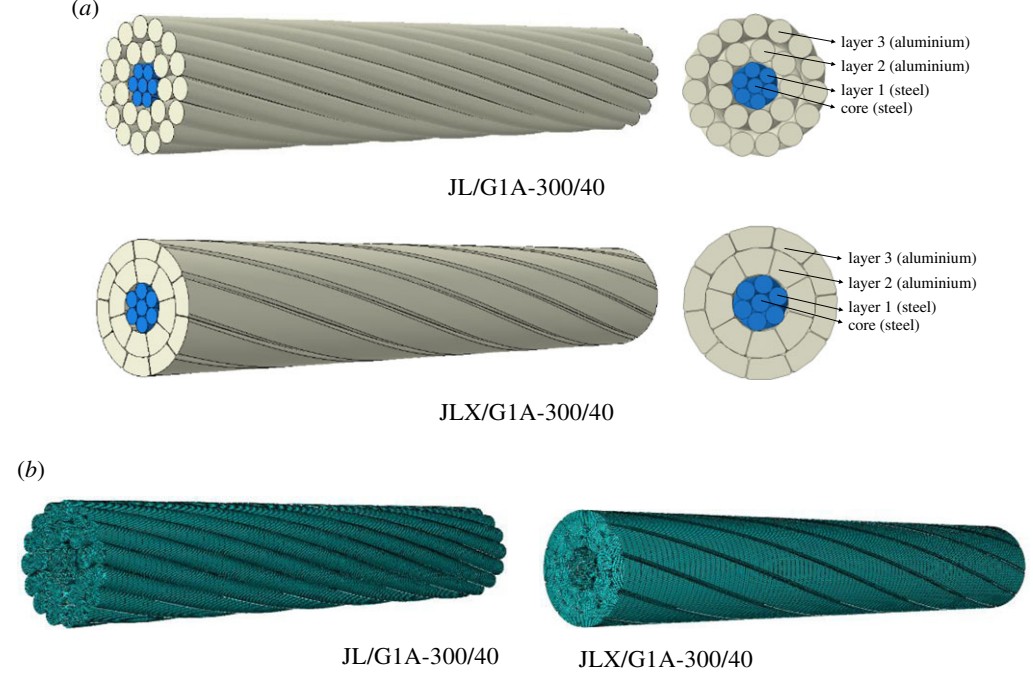

**Figure 5.** Geometrical and FE models of two conductors. (*a*) Geometrical model; (*b*) FE model.

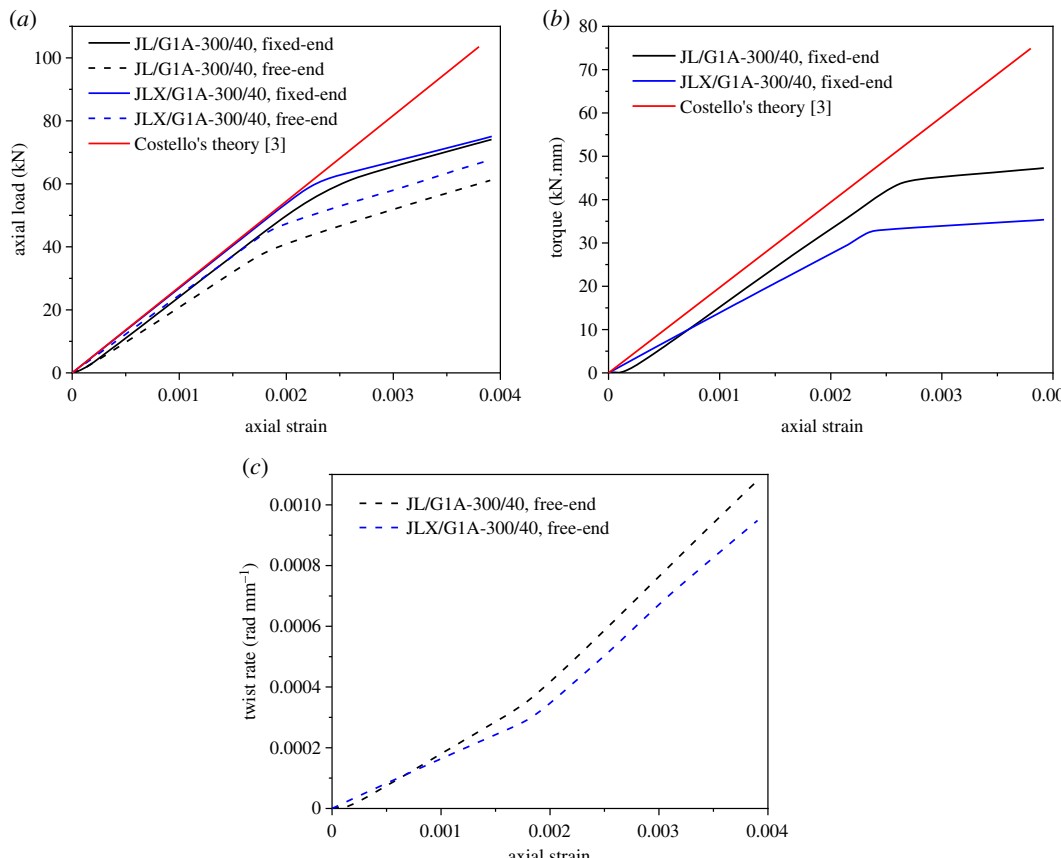

**Figure 6.** Overall tensile behaviour of the two conductors. (*a*) Relationship between axial strain and axial load; (*b*) relationship between axial strain and torque; (*c*) relationship between axial strain and twist rate.

Nonlinearity of the curves determined by the numerical simulations is obvious but only linear relationship is predicted by Costello's theory. At the beginning of loading, nonlinearity of the curves for conductor JL/G1A-300/40 is observed, for which the gaps between the round wires may be

**Table 4.** Stiffness coefficients of the two conductors.

| conductor | $k_{\varepsilon\varepsilon}$ (kN) | $k_{\theta\theta}$ (kN mm$^2$) | $k_{\varepsilon\theta}$ (kN mm) | $k_{\theta\varepsilon}$ (kN mm) |
|---|---|---|---|---|
| JL/G1A-300/40 | $2.6 \times 10^4$ | 86 000 | $1.9 \times 10^4$ | $1.8 \times 10^4$ |
| JLX/G1A-300/40 | $2.6 \times 10^4$ | 92 000 | $1.0 \times 10^4$ | $1.4 \times 10^4$ |

responsible, and nearly no nonlinearity for conductor JLX/G1A-300/40 occurs because there are nearly no gaps between aluminium wires as shown in figure 5a. When the axial strain reaches approximately 0.002, the four loading curves of axial load and axial strain enter plastic stage and their slopes decrease slightly as shown in figure 6a. Moreover, the loads in the fixed-end case for the two conductors are larger than those in the free-end case corresponding to the same axial strain, which is consistent with the conclusion proposed by Judge *et al.* [11]. It is noted that Young's modulus determined by the numerical simulation for the two conductors in the fixed-end case is 5.4% larger than that provided in the standard [29].

If the end rotation of a conductor segment around its axis is fixed, torque will occur at the end during loading. From the relationships between the axial strain and torque for the two conductors in the fixed-end case as shown in figure 6b, it is seen that there is a slight nonlinearity of the curve for conductor JL/G1A-300/40 at the beginning, which is due to the gaps between the wires. With the increase of the axial load, the gaps between wires firstly are filled and torque generates in the conductor, which increases and becomes larger than that in conductor JLX/G1A-300/40 in the case of larger deformation.

If the end rotation of a conductor segment is free, the conductor will twist around its axis during applying axial tensile load. The relationships between the axial strain and twist rate for the two conductors in the free-end case are shown in figure 6c. Because of the gaps between the wires, nonlinearity at the initial stage of conductor JL/G1A-300/40 occurs too. Moreover, the twist rate of JLX/G1A-300/40 is lower than that of JL/G1A-300/40 corresponding to the same axial strain in the case of larger deformation.

### 3.3.2. Coupling of tension and torsion deformations of conductors

As discussed in the previous section, twist takes place for the two conductors in free-end case although only axial tensile load is applied to the end of the segment, and this indicates that there is coupling between tension and torsion deformations due to the helical symmetry of the conductors. The overall elastic behaviour of the conductor can be expressed in the form [31]

$$\begin{bmatrix} F_z \\ M_z \end{bmatrix} = \begin{bmatrix} k_{\varepsilon\varepsilon} & k_{\varepsilon\theta} \\ k_{\theta\varepsilon} & k_{\theta\theta} \end{bmatrix} \begin{bmatrix} \varepsilon_z \\ \theta_z \end{bmatrix}, \tag{3.1}$$

where $F_z$ is the axial load, $M_z$ is the torque, $\varepsilon_z$ is the overall axial strain, $\theta_z$ is the twist rate defined as twist angle per unit length, and $k_{\varepsilon\varepsilon}$, $k_{\theta\theta}$, $k_{\theta\varepsilon}$ and $k_{\varepsilon\theta}$ are the so-called stiffness coefficients corresponding to pure tension, torsion and coupling deformation. In the fixed-end case, when tensile load $F_z$ is applied, the torsional rate $\theta_z = 0$ and there is a torque $M_z$ at the coupled point. According to equation (3.1), it is obtained that $k_{\varepsilon\varepsilon} = F_z / \varepsilon_z$ and $k_{\theta\varepsilon} = M_z / \varepsilon_z$. On the other hand, in the free-end case, when tensile load $F_z$ is applied, there is a torsional rate $\theta_z$, and the torque $M_z$ at the coupled point is zero. Similarly, it is obtained that $k_{\varepsilon\theta} = (F_z - k_{\varepsilon\varepsilon} \cdot \varepsilon_z) / \theta_z$ and $k_{\theta\theta} = k_{\theta\varepsilon} \cdot \varepsilon_z / \theta_z$ according to equation (3.1). Based on the numerical simulation results, the stiffness coefficients of the two conductors are obtained and listed in table 4, from which it is known that the tension–torsion coupling effect of conductor JLX/G1A-300/40 is less than that of JL/G1A-300/40 in the case of larger deformation.

### 3.3.3. Mises stress distributions of conductors

The Mises stress distributions of conductors JL/G1A-300/40 and JLX/G1A-300/40 under 30 kN tension in fixed-end and free-end cases are shown in figures 7 and 8, from which it is seen that all the steel wires and aluminium wires in layer 3 of the two conductors in the two cases are in the elastic range, but partial regions of the aluminium wires in layer 2 of JL/G1A-300/40 exceed the elastic range, and plastic deformation takes place. The maximum Mises stresses of the steel wires of conductor JL/G1A-300/40 in both cases are larger than those of the steel wires of JLX/G1A-300/40. For both conductors, the maximum stresses of the steel wires in both cases are close to each other. On the other hand, the

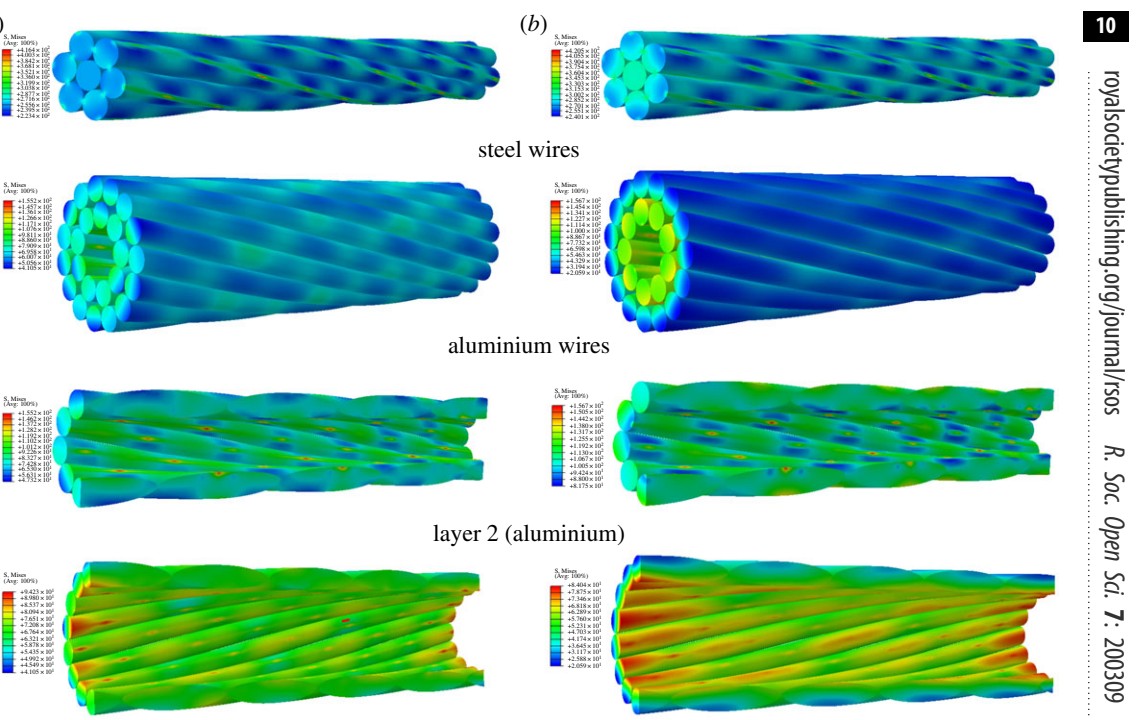

steel wires

aluminium wires

layer 2 (aluminium)

layer 3 (aluminium)

**Figure 7.** Stress distribution of JL/G1A-300/40 conductor under 30 kN tension. (*a*) Stress distribution in fixed-end case; (*b*) stress distribution in free-end case.

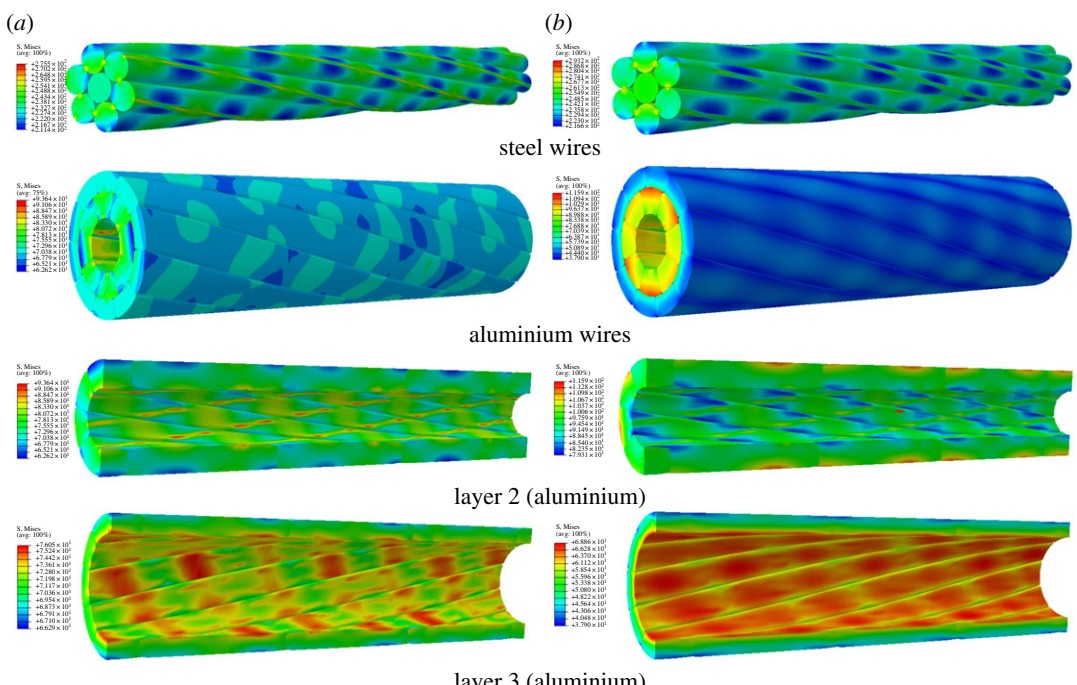

steel wires

aluminium wires

layer 2 (aluminium)

layer 3 (aluminium)

**Figure 8.** Stress distribution of JLX/G1A-300/40 conductor under 30 kN tension. (*a*) Stress distribution in fixed-end case; (*b*) stress distribution in free-end case.

maximum stresses of the aluminium wires of conductor JL/G1A-300/40 are apparently larger than those of conductor JLX/G1A-300/40 in both cases. The round shape of the aluminium wires of the former conductor leading to line–line contact between the wires may be responsible for the larger stress concentration. Due to the flat cross-section shape of the aluminium wires in conductor JLX/G1A-300/40, the maximum stresses between the wires are lower. For both conductors, the largest stress occurs in

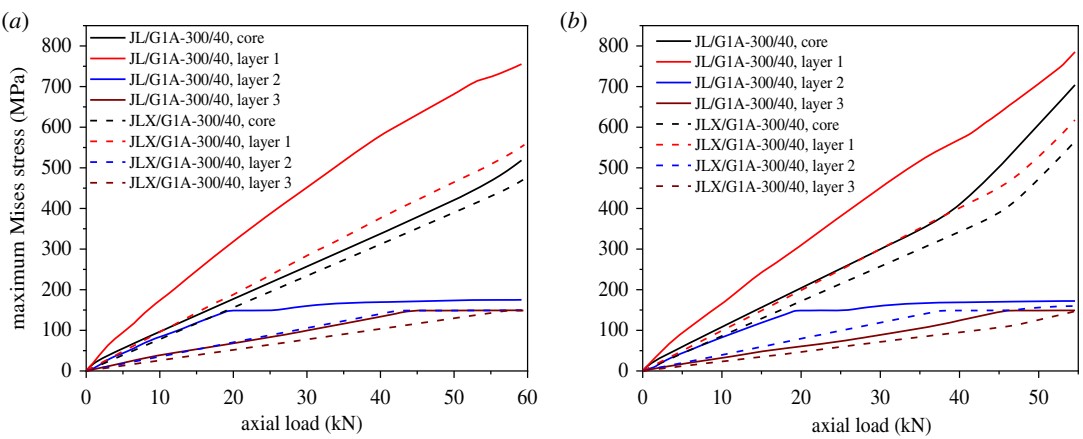

**Figure 9.** Relations between maximum Mises stress in each layer of two conductors and axial load. (*a*) Relationships in fixed-end case; (*b*) relationships in free-end case.

the steel wires of layer 1 and the smallest stress occurs in the aluminium wires in layer 3, which indicates that the load on a conductor is mainly sustained by the steel wires. Finally, the maximum stresses of steel wires and aluminium wires in layer 2 of the two conductors in the free-end case are larger than those in the fixed-end case, which is consistent with the result obtained by Xiang *et al.* [32].

In addition, the maximum Mises stress in each layer of the two conductors during tensile loading in both cases is shown in figure 9. It is obvious that the maximum stress in each layer of conductor JL/G1A-300/40 is larger than those of the corresponding layer of conductor JLX/G1A-300/40 in both cases. The largest maximum stresses in the steel wires of JL/G1A-300/40 are larger by 37% and 44% than those of JLX/G1A-300/40 in fixed-end case and free-end case, respectively, and the largest stresses in the aluminium wires of the former conductor are larger by 111% and 97% than those of the latter one. When the axial load reaches 33 kN, the maximum stress in aluminium wires of conductor JL/G1A-300/40 reaches ultimate stress, but the stresses of conductor JLX/G1A-300/40 are still in elastic range.

Based on the above discussion, it is known that the stresses in steel wires of the core and layer 1 are much larger than those in the aluminium wires of layers 2 and 3 when the conductors are subjected to the same load, which indicates that the load of a transmission conductor line is sustained mainly by the steel wires. Moreover, the maximum stress in layer 1 is larger than that in the core at the same axial load, and the maximum stress in layer 2 is larger than that in layer 3. These conclusions may provide instructions for the design of a conductor with new structure.

It is known from figures 7 and 8 that the plastic deformation occurs in aluminium wires in layer 2 of conductor JL/G1A-300/40 subjected to tension of 30 kN, and the equivalent plastic strain distribution in the aluminium wires is shown in figure 10. It is seen that the plastic strain of the aluminium wires occurs on the inner faces contacting with the steel wires, and the plastic strains in the fixed-end case are lower than those in free-end case.

### 3.3.4. Contact pressure distributions of conductors

In the 3D FE models of the conductors, contact condition between the wires is set, so the aluminium wires slip to each other when the conductors are under stretching. Once slipping between two wires takes place, the contact between them will inevitably induce wear [33], and the contact pressure between a pair of contact surfaces is a key parameter leading to wear of the wires. Contact pressure distributions on inner surfaces of layer 2 when the conductors are under 30 kN tension in both cases are shown in figure 11, and the maximum pressures of the two conductors during applying loading in the two cases are shown in figure 12. It can be seen that the maximum contact pressures in both conductors increase nonlinearly with the axial loads, and the maximum contact pressure in each layer in conductor JLX/G1A-300/40 is lower than that of the corresponding layer in conductor JL/G1A-300/40, which indicates that under the same load the wear of the former conductor is less than that of the latter one. The round cross-section of the aluminium wires in JL/G1A-300/40 leads to point-to-point contact and line-to-line contact between the wires and it gives rise to larger contact stress. On the other hand, the flat cross-section of the aluminium wires in JLX/G1A-300/40 leads to line-to-line contact and surface-to-surface contact between the wires, so the contact stress is lower.

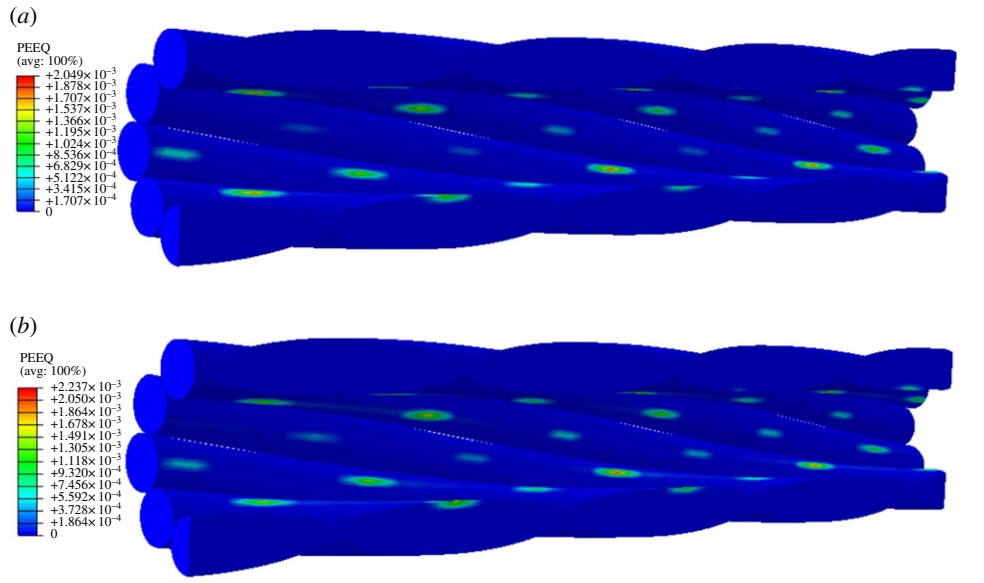

**Figure 10.** Equivalent plastic strain of aluminium wires in layer 2 of conductor JL/G1A-300/40 under 30 kN tension. (*a*) Plastic strain in aluminium wires in fixed-end case; (*b*) plastic strain in aluminium wires in free-end case.

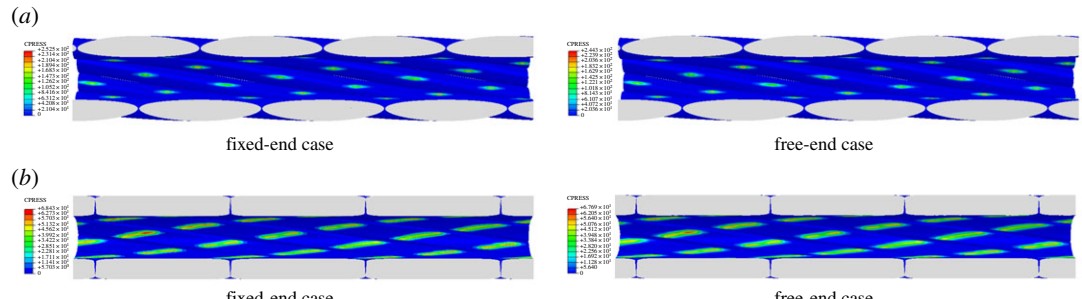

**Figure 11.** Contact pressure on inner faces of layer 2 as conductors under 30 kN tension. (*a*) Contact pressure of conductor JL/G1A-300/40; (*b*) contact pressure of conductor JLX/G1A-300/40.

It is also found that the maximum contact pressures of the two conductors in the fixed-end case are larger than those in the free-end loading case. At the same axial load, the maximum contact pressure in layer 1 is larger than that in the layer 2, and the maximum contact pressure in layer 2 is larger than that in layer 3 for the two conductors.

# 4. Dynamic characteristics of conductors

There are a lot of numerical simulation works on the dynamic responses of transmission lines under different dynamic loads such as ice-shedding and wind loads [23–27]. In these works, the conductors are simulated by cable or beam elements, which cannot reflect the real stress in the wires of the stranded conductors due to ignoring their structural details. The stress distributions estimated by the beam elements, which may be much lower than those by the 3D refined models, are used for the safety design. As we know, many incidences of conductor breakage took place in transmission lines even though the lines were safe based on the design criterion of the traditional method in which the structural details of the conductors are ignored. Therefore, it is important to investigate the stress and deformation of the steel and aluminium wires of the stranded conductors during vibration under dynamic loads, which may provide data for the strength assessment and fatigue life estimation of the conductors.

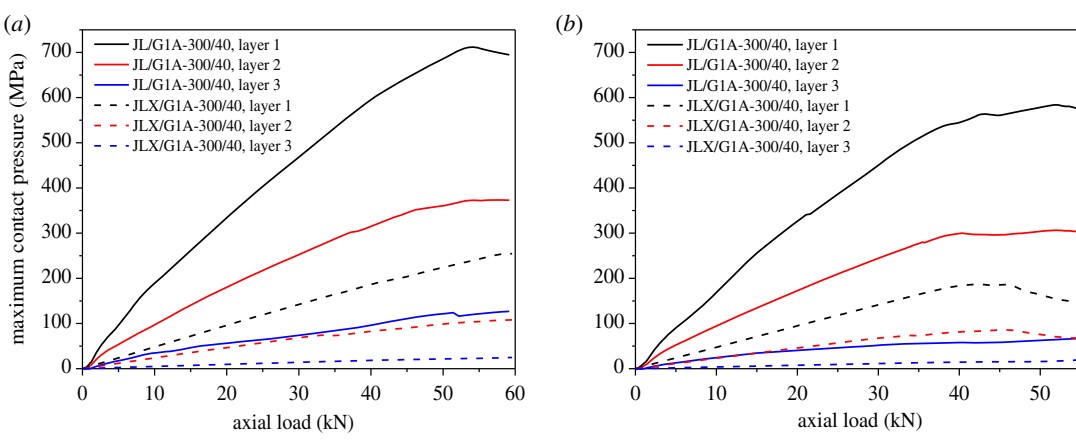

**Figure 12.** Maximum contact pressures of conductors during loading. (*a*) Maximum contact pressure in fixed-end case; (*b*) maximum contact pressure in free-end case.

**Table 5.** Parameters of conductors and insulator strings.

| component | Young's modulus (MPa) | Poisson's ratio | equivalent radius (mm) | density (kg m$^{-3}$) |
|---|---|---|---|---|
| conductor | 73 000 | 0.3 | 10.3881 | 3342 |
| insulator string | 21 000 | 0.3 | 40 | 9182 |

## 4.1. FE modelling method of a transmission line after ice-shedding

Two typical three-span single conductor lines with equal 300 m span length and initial tension of 37.86 MPa are studied. The conductors of the two lines are respectively JL/G1A-300/40 and JLX/ G1A-300/40. The length of the suspension insulant string is 5.2 m. The parameters of the conductors and insulator strings are as listed in table 5. It is assumed that ice with total mass of 2234 kg is accreted on the two lines. The equivalent thickness of the ice on conductors JL/G1A-300/40 is 20 mm and that of conductor JLX/G1A-300/40 is 20.74 mm. The case that the ice sheds from the middle span with 100% ice-shedding rate is numerically simulated.

Firstly, the dynamic responses of the two lines after ice-shedding are numerically simulated by means of the method presented by Yan *et al.* [24], in which conductor is simulated by beam elements. Based on the simulation results, it is known that the dangerous region of the central span is located in the vicinity of the suspensions, so a 143.5 mm long segment of conductor near the suspension originally simulated by beam elements is replaced by a refined 3D solid model and the new FE models, known as mixed FE models. In the mixed model, each end surface of the solid element model is coupled with a reference point which is connected with the beam node using the kinematic coupling type. Thus, the beam node and the coupling point of solid model have the same displacements. The mixed FE models discretized by 3D elements and beam elements are shown in figure 13. The dynamic responses of the two conductor lines are then simulated by the mixed FE models again.

The vertical displacement and tension time histories at middle point of span two of the beam FE models and the mixed models are shown in figure 14, from which it is seen that the dynamic responses at the middle point of span two by the two kinds of models are close to each other. In the meantime, it is known that the frequencies of the mixed model and the beam model are nearly the same and the relative difference is about 3%, through spectral analysis for the tension time history curves. However, the stress distributions in the dangerous regions of the two conductors are revealed by the local 3D models in mixed models, which is very important for the fatigue life evaluation of the wires.

## 4.2. Dynamic responses of conductors with structural details

Although the displacement and tension time histories at the middle point of span two of the conductors determined by the beam models and the mixed models are very close, the maximum stresses at the

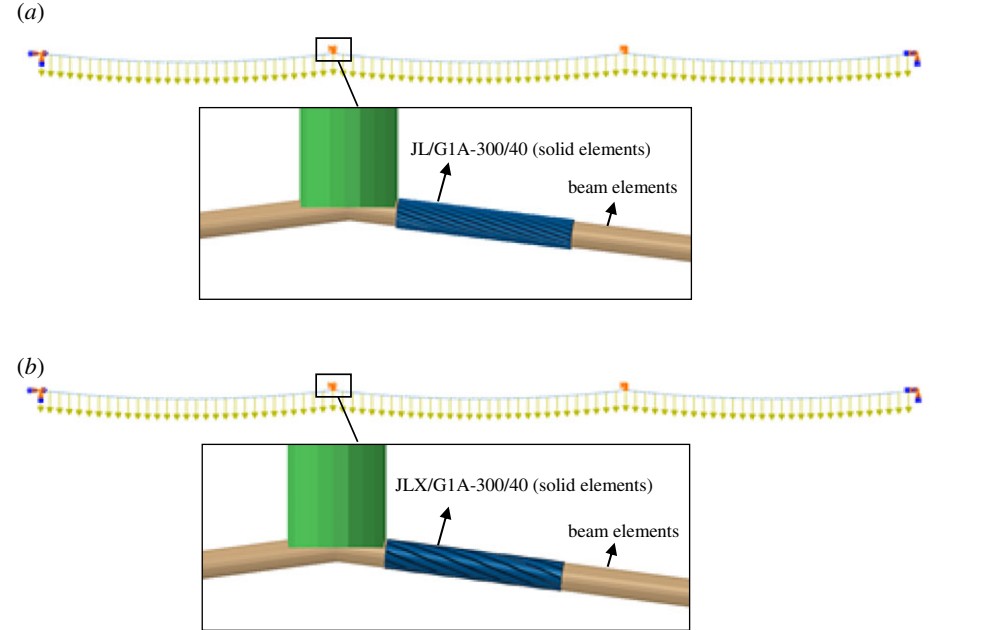

**Figure 13.** Mixed FE models of three-span conductor lines. (*a*) Mixed FE model of conductor JL/G1A-300/40; (*b*) mixed FE model of conductor JLX/G1A-300/40.

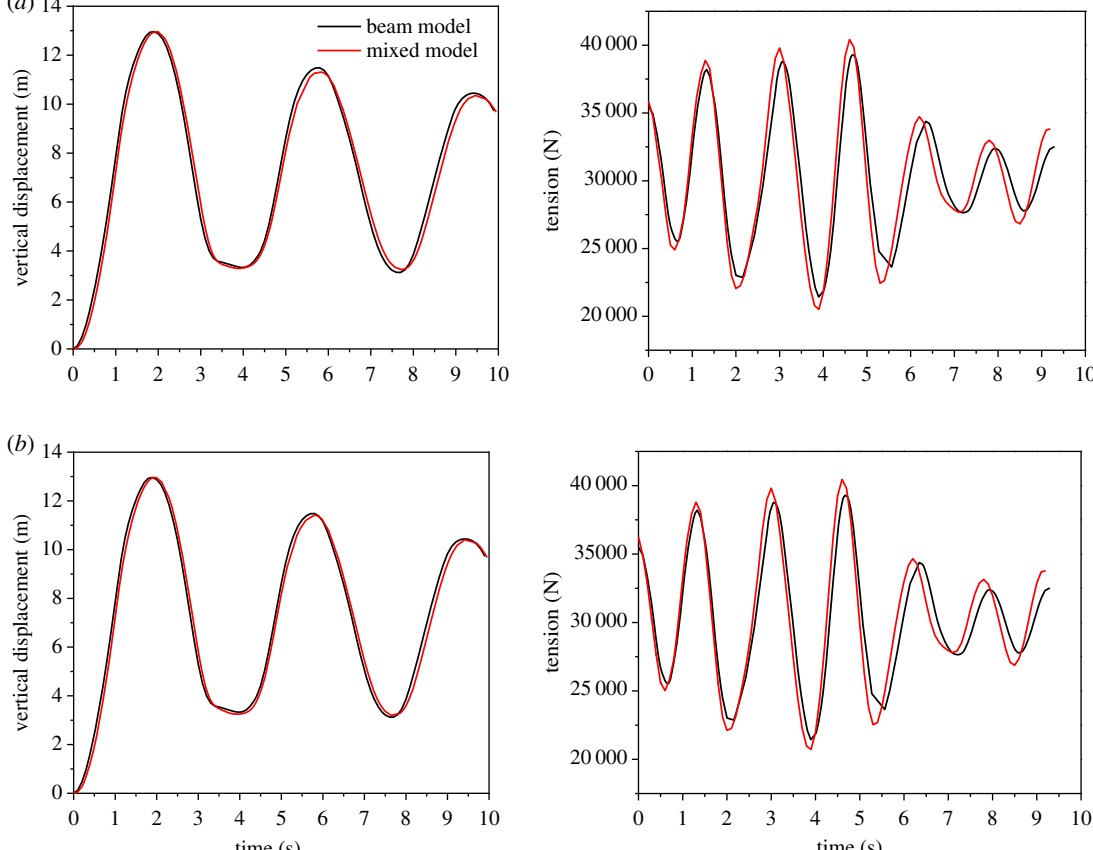

**Figure 14.** Vertical displacement and tension time histories at the middle point of span 2 after ice-shedding. (*a*) Vertical displacement and tension of conductor JL/G1A-300/40; (*b*) vertical displacement and tension of conductor JLX/G1A-300/40.

dangerous region of the conductors by the beam model and the mixed models are very different as shown in figure 15. It is shown that the maximum stresses in different layers of the conductors obtained by the mixed FE models are very different, and the maximum stresses by the beam FE

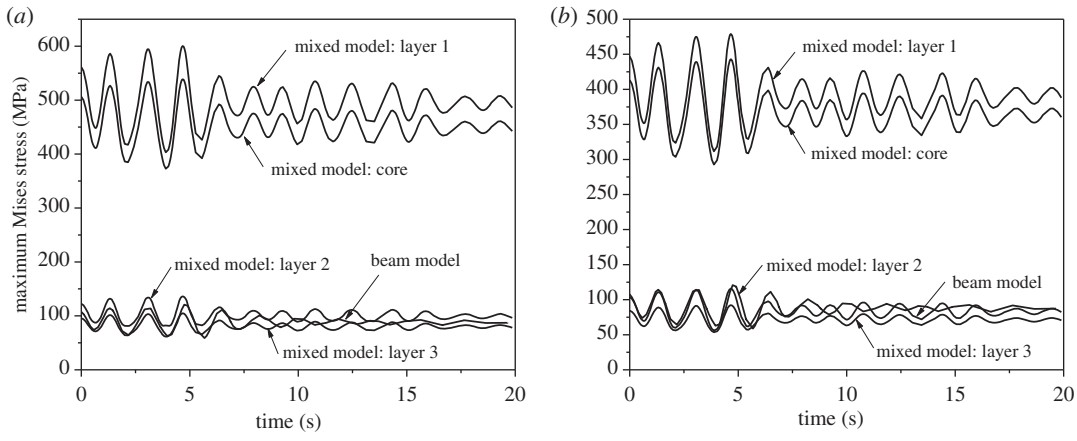

**Figure 15.** Maximum stress time histories in the dangerous region by beam models and mixed models. (*a*) Maximum Mises stress of conductor JL/G1A-300/40; (*b*) maximum Mises stress of conductor JLX/G1A-300/40.

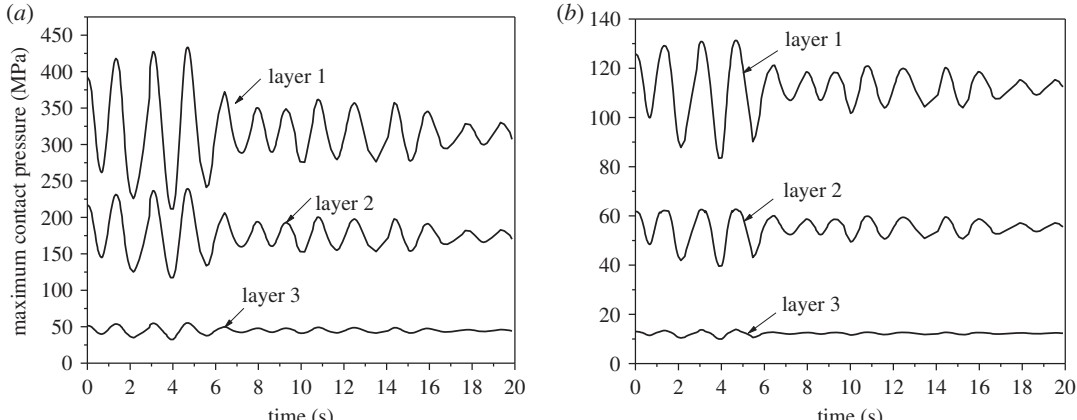

**Figure 16.** Maximum contact pressure time histories of conductors after ice-shedding. (*a*) Maximum contact pressure of conductor JL/G1A-300/40; (*b*) maximum contact pressure of conductor JLX/G1A-300/40.

models are much lower than those by the mixed models, which implicates that real maximum stress cannot be revealed by the beam element model and local 3D simulation of conductors with strand structure details is necessary.

All the maximum stress time history curves of the two conductors shown in figure 15 arrive at peaks at 4.67s after ice-shedding. The maximum stress of the aluminium wires (layer 2) of the conductors JL/G1A-300/40 and JLX/G1A-300/40 determined by the mixed models are, respectively, 136 and 116 MPa, and those of the steel wires (layer 1) are, respectively, 600 and 479 MPa for the two conductors. The maximum stress in each layer of conductor JL/G1A-300/40 is larger than that of the corresponding layer of conductor JLX/G1A-300/40. Both the maximum stresses in aluminium and steel wires are smaller than the material strengths of aluminium and steel, respectively, and this means that no abrupt breakage takes place in this ice-shedding case. However, the dynamic stress response may lead to fatigue failure. The maximum Mises stress in layer 2 is bigger than that in layer 3. It means that the first wire failure will occur at the inner layer, which is consistent with the experimental and numerical works on Crow ACSR [21].

The maximum contact pressure time histories of steel and aluminium wires of the two conductors after ice-shedding are shown in figure 16. It is seen that the time histories of the three layers of wires are very different and all the maximum values arrive at peaks at 4.67 s after ice-shedding. The maximum contact pressures of layers 1, 2 and 3 of conductor JL/G1A-300/40 are, respectively, 433, 239 and 55 MPa, and those of conductor JLX/G1A-300/40 are, respectively, 131, 63 and 14 MPa. The contact pressure will affect the wear of the wires during vibration.

It is noted that the maximum Mises stress and the maximum contact pressure are analysed only for the solid elements far enough away from the connection between the beam conductor model and the

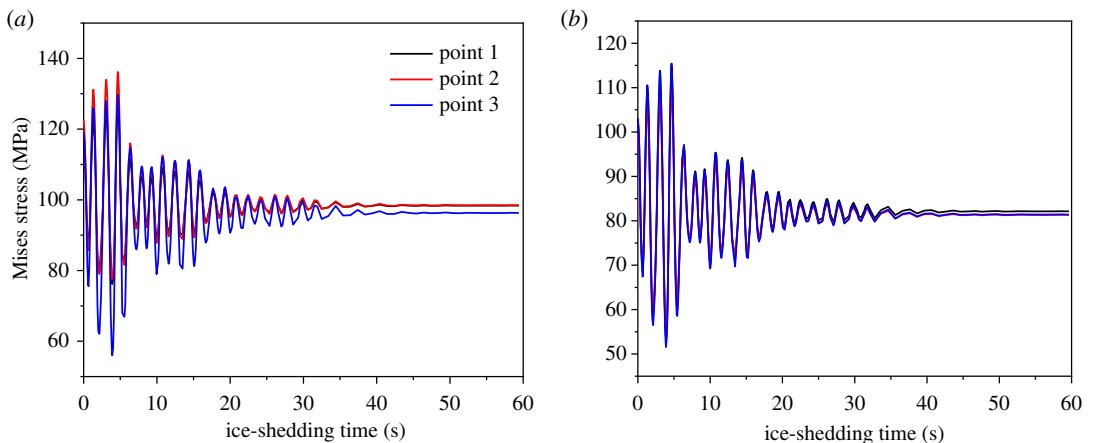

**Figure 17.** Stress time histories of typical points of aluminium wires in layer 2. (*a*) Stress time histories at typical points of JL/G1A-300/40; (*b*) stress time histories at typical points of JLX/G1A-300/40.

solid conductor model. Therefore, the contact pressure distribution of this part is less affected by the connection between the two models.

## 4.3. Fatigue and wear of conductors

Although the maximum stresses in the wires of the conductors usually are less than the material strengths of the conductors under dynamic loads, fatigue failure and wear may take place under the repeated dynamic loads such as ice-shedding. As we know, most of the conductor breakages are induced by fatigue and wear in the wires of the conductors during operation.

It is known that the electric utility and cable manufacturers usually only perform standard vibration and fatigue tests to verify whether the conductors meet the factory standard. Ice-shedding and wind loads on overhead line conductors are usually complicated, and it is difficult to simulate in the laboratory. On the other hand, the fatigue tests are very time-consuming and costly. So, it is important to use detailed numerical models to analyse the fatigue lives of conductors in service.

Based on the numerical simulation of the dynamic responses of the transmission lines with conductors JL/G1A-300/40 and JLX/G1A-300/40 after ice-shedding, it is known that the stresses in the inner aluminium (layer 2) wires are larger than those of aluminium wires in the other layer, so the stress time histories of some critical points in the inner aluminium wires are selected for fatigue life estimation. The stress histories at 20 typical points of the conductors after ice-shedding are extracted from the simulation results, and the Mises stress time histories of three points in layer 2 of the two mixed models are shown in figure 17.

It is suggested by Lalonde *et al.* [21] that the Basquin equation should be adopted to describe the relation between stress amplitude and fatigue life of an aluminium wire, and the Basquin equation is the following:

$$\sigma_a = \sigma'_f (2N)^b, \tag{4.1}$$

where $\sigma_a$ is the stress amplitude, $N$ is the corresponding fatigue life, $\sigma'_f$ and $b$ are fatigue parameters of the material. The $\sigma'_f$ and $b$ of 1350-H19 aluminium used for ACSR are, respectively, 204 MPa and −0.07 [21], and these parameters are used to estimate the fatigue lives of the aluminium wires of the two types of conductors discussed in this paper.

As shown is figure 17, the dynamic stresses of the wires vary with time after ice-shedding. The Miner's criterion is employed to estimate the fatigue lives of the wires. According to the Miner linear cumulative damage theory [34], the total damage can be figured out by

$$D = \frac{\sum_1^K n_i}{N_i}, \tag{4.2}$$

where $D$ is the total damage, $N_i$ is the fatigue life corresponding to a specific stress amplitude $\sigma_a^i$ ($i = 1 \sim K$), $n_i$ is cycle number of the applied stress amplitude $\sigma_a^j$, and $K$ is the number of the stress amplitude level. When $D$ arrives at 1.0, fatigue failure takes place.

To estimate the fatigue life, the fatigue life $N_i$ corresponding to each stress amplitude $\sigma_a^i$ of a typical point in the wires during vibration of the conductor after ice-shedding is firstly determined by equation (4.1), and the stress amplitude $\sigma_a^i$ as well as the cycle number $n_i$ corresponding to stress amplitude $\sigma_a^i$ are calculated by the rainflow counting method based on the stress history curves. Then the damage parameter $D$ after one ice-shedding case is obtained by summation of all the damage parameters corresponding to each stress cycle of the time history curves. Fatigue failure will take place when $D = 1$, which means fatigue failure of an aluminium wire may take place after $1/D$ times of ice-shedding.

With the method discussed above, the damage parameters of the 20 typical points in the aluminium wires of the two conductors are calculated and the maximum values are $D = 9.62 \times 10^{-7}$ for conductor JL/G1A-300/40 and $1.24 \times 10^{-7}$ for conductor JLX/G1A-300/40 after one ice-shedding case. Therefore, fatigue failure of the aluminium wires of the two conductors may take place after $1.04 \times 10^{6}$ times of ice-shedding for conductor JL/G1A-300/40 and $8.06 \times 10^{6}$ times for conductor JLX/G1A-300/40. It is noted that the ice-shedding case discussed in this paper is assumed and may be different from a real case, so the estimated fatigue lives of the wires may be different from the real situation. However, the presented method could be used to estimate fatigue lives of the conductors under any dynamic loading. The evaluation of fatigue lives of conductors in service can provide references for the transmission line engineering.

On the other hand, it is difficult to assess the wear between the wires due to the complicated structures of the conductors and the sophisticated loading conditions of the transmission lines in operation. Some researchers [35–37] have investigated the effect of parameters, such as contact load, crossing angle and torsion angle, on wear performance of steel strands by means of experiments. Wear coefficient is defined as the amount of wear per unit of wear stroke and per unit wear load, which may be used to indicate the wear resistance of the material. It is obtained in [37] that the relationship between the wear coefficient and average contact pressure of steel wires is approximately linear. The larger the contact pressure, the larger the wear coefficient. It is known from §§3.3.4 and 4.2 that the contact pressures between the wires of conductor JL/G1A-300/40 are larger than those of conductor JLX/G1A-300/40 under the same conditions, so the wear of the former is more severe than that of the latter one. Unfortunately, there is no experimental data of the wear coefficients of steel–aluminium and aluminium–aluminium of the conductors, which needs further investigation.

# 5. Conclusion

In this paper, the refined elastoplastic 3D FE models of both ACSR and FACSR conductors with structural details are set up. And the high accuracy of the models is validated with existing experimental and numerical data of steel cables subjected to quasi-static loading. Using the numerical models, the elastoplastic deformation and stress distributions of the two types of conductors under static loads are investigated. Furthermore, using the mixed models, in which the transmission conductors are discretized with refined 3D models and equivalent beam models, the dynamic responses of these transmission lines after ice-shedding are in turn analysed. It is concluded that:

(i) The stresses in the steel and aluminium wires are different from those determined by means of the equivalent conductor models, so it is necessary to take into account the effect of the structural details of the conductors on their mechanical behaviour in the design of transmission lines.

(ii) The maximum Mises stresses in the layers of conductors are ranked in descending order as layer 1, the core, layer 2 and layer 3 in static axial loading and dynamic ice-shedding situations.

(iii) The tensile stiffness of the two types of conductors is close to each other, but the coupling effect between tension and torsion of ACSR is more obvious than that of FACSR in the case of larger deformation.

(iv) Under the same tensile load, the maximum stress in the wires of ACSR is much larger than that of the wires of FACSR, which indicates the safety of the latter one is higher than that of the former under the same load.

(v) Based on the numerical simulation of the dynamic responses of the transmission lines with the two types of conductors after ice-shedding, it is known that the estimated fatigue life of FACSR is longer than that of ACSR under the same dynamic load.

Data accessibility. Data available from the Dryad Digital Repository: https://doi.org/10.5061/dryad.c2fqz614q [38].

Authors' contributions. J.L., B.Y. and G.H. carried out the tensile tests, participated in the static and dynamic numerical simulation analysis on conductors, participated in the design of the study and drafted the manuscript; Z.M., X.L. and H.Z. participated in the analysis of the fatigue and wear of conductors, and participated in data analysis. All authors gave final approval for publication.

Competing interests. We declare we have no competing interests.

Funding. Financial support came from the National Natural Science Foundation of China (grant no. 11572060).

Acknowledgements. We thank professor Yuanqing Li, teacher Hao Liu and teacher Shuyan Nie for their contributions to the tensile tests, Chuan Wu for his providing the transmission conductors, Huachao Deng and Nan Wen for their help in the numerical simulation and Cheng Hou for his advice on the details of this paper. Besides, we also thank editors and anonymous reviewers for their helpful suggestions on the early versions of this manuscript.

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
