## [Reviewer comments · Royal Society Open Science]

Review History

RSOS-200309.R0 (Original submission)

Review form: Reviewer 1 (Xiaohui Liu)

Is the manuscript scientifically sound in its present form?

Yes

Are the interpretations and conclusions justified by the results?

Yes

Is the language acceptable?

Yes

Do you have any ethical concerns with this paper?

No

Have you any concerns about statistical analyses in this paper?

No

Recommendation?

Accept with minor revision (please list in comments)

Comments to the Author(s)

See attached Files (Appendix A).

Review form: Reviewer 2

Is the manuscript scientifically sound in its present form?

Yes

Are the interpretations and conclusions justified by the results?

Yes

Is the language acceptable?

Yes

Do you have any ethical concerns with this paper?

No

Have you any concerns about statistical analyses in this paper?

No

Recommendation?

Accept with minor revision (please list in comments)

Comments to the Author(s)

The authors present the results of detailed finite element analysis studies of stranded conductors in both static and dynamic regimes of loading. They also compare the response of ACSR conductors and FACSR conductors and their results indicate that fatigue life of the latter is longer: considering that fatigue life can be tested experimentally (quite routinely by electric utility and cable manufacturers) in a reliable fashion it is questionable whether high fidelity numerical models are needed for this purpose.

Given the many uncertainties and simplifications used in the ice shedding studies, one would also question the need for more refined cable models to predict the response in terms of cable jumps and wire tensions. Several recent publications on ice shedding from overhead line conductors have addressed such assumptions and important parameters. It does not mean that detailed modelling is wrong, it perhaps is simply not necessary to give additional accuracy to a problem already posed with very inaccurate loading conditions.

In section 1, the authors have ignored two very relevant bodies of work done by Masoud Roshan Fekr et al. and Gang Qi and McClure at McGill University. It seems that the authors have reviewed Roshan Fekr's work on ice shedding (ref. 19) but have not pursued their review of his later work on stranded cable modelling.

See:

Roshan Fekr, M., McClure, G., and Farzaneh, M. 1999. Application of ADINA to stress analysis of an optical ground wire. *Computers and Structures*, 72: 301-316.

PhD thesis by Masoud Roshan Fekr: Stress analysis of an optical ground wire. 1999:

<https://mcgill.on.worldcat.org/detailed-record/898040899?databaseList=283&databaseList=638&scope=wz:12129>

Gang Qi's modelling work is more recent and also more advanced. It has not been published widely (only in conferences):

Qi, G., and McClure, G. 2013. Advanced computational stress analysis of a stranded overhead line conductor under fretting fatigue conditions. *Proc. 3rd Specialty Conf. On Materials Engineering and Applied Mechanics, CSCE Annual Conference, Montréal, Canada, 29 May – 1 June. Paper MEC-067.*

Qi, G., McClure, G., and Roshan Fekr, M. 2011. Stress Analysis of an Optical Ground Wire using Finite Elements. Abstract (P. 115) and oral presentation at the Sixth M.I.T. Conference on Computational Fluid and Solid Mechanics, Cambridge, MA, USA, 15-17 June.
 PhD thesis by Gang Qi: Computational modeling for stress analysis of overhead transmission line stranded conductors under design and fretting fatigue conditions:
<https://mcgill.on.worldcat.org/detailed-record/880373447?databaseList=283&databaseList=638&scope=wz:12129>

It is recommended that the authors review these works and indicate how their own studies differ and or agree. At the very least there should be proper citations.

Decision letter (RSOS-200309.R0)

31-Mar-2020

Dear Dr Liu

On behalf of the Editors, I am pleased to inform you that your Manuscript RSOS-200309 entitled "Study on mechanical characteristics of conductors with 3D finite element models" has been accepted for publication in Royal Society Open Science subject to minor revision in accordance with the referee suggestions. Please find the referees' comments at the end of this email.

The reviewers and handling editors have recommended publication, but also suggest some minor revisions to your manuscript. Therefore, I invite you to respond to the comments and revise your manuscript.

- Ethics statement

- Data accessibility

If you wish to submit your supporting data or code to Dryad (<http://datadryad.org/>), or modify your current submission to dryad, please use the following link:
<http://datadryad.org/submit?journalID=RSOS&manu=RSOS-200309>

- Competing interests

- Authors' contributions

- Acknowledgements

- Funding statement

Because the schedule for publication is very tight, it is a condition of publication that you submit the revised version of your manuscript before 09-Apr-2020. Please note that the revision deadline will expire at 00.00am on this date. If you do not think you will be able to meet this date please let me know immediately.

- 1) A text file of the manuscript (tex, txt, rtf, docx or doc), references, tables (including captions) and figure captions. Do not upload a PDF as your "Main Document";

- 2) A separate electronic file of each figure (EPS or print-quality PDF preferred (either format should be produced directly from original creation package), or original software format);
- 3) Included a 100 word media summary of your paper when requested at submission. Please ensure you have entered correct contact details (email, institution and telephone) in your user account;
- 4) Included the raw data to support the claims made in your paper. You can either include your data as electronic supplementary material or upload to a repository and include the relevant doi within your manuscript. Make sure it is clear in your data accessibility statement how the data can be accessed;
- 5) All supplementary materials accompanying an accepted article will be treated as in their final form. Note that the Royal Society will neither edit nor typeset supplementary material and it will be hosted as provided. Please ensure that the supplementary material includes the paper details where possible (authors, article title, journal name).

If your manuscript is newly submitted and subsequently accepted for publication, you will be asked to pay the article processing charge, unless you request a waiver and this is approved by Royal Society Publishing. You can find out more about the charges at <https://royalsocietypublishing.org/rsos/charges>. Should you have any queries, please contact openscience@royalsociety.org.

on behalf of Dr Derek Abbott (Associate Editor) and R. Kerry Rowe (Subject Editor)
openscience@royalsociety.org

Reviewer comments to Author:
Reviewer: 1

Comments to the Author(s)
See attached Files

Reviewer: 2

Comments to the Author(s)

The authors present the results of detailed finite element analysis studies of stranded conductors in both static and dynamic regimes of loading. They also compare the response of ACSR conductors and FACSR conductors and their results indicate that fatigue life of the latter is longer: considering that fatigue life can be tested experimentally (quite routinely by electric utility and cable manufacturers) in a reliable fashion it is questionable whether high fidelity numerical models are needed for this purpose.

Given the many uncertainties and simplifications used in the ice shedding studies, one would also question the need for more refined cable models to predict the response in terms of cable jumps and wire tensions. Several recent publications on ice shedding from overhead line conductors have addressed such assumptions and important parameters. It does not mean that detailed modelling is wrong, it perhaps is simply not necessary to give additional accuracy to a problem already posed with very inaccurate loading conditions.

In section 1, the authors have ignored two very relevant bodies of work done by Masoud Roshan Fekr et al. and Gang Qi and McClure at McGill University. It seems that the authors have reviewed Roshan Fekr's work on ice shedding (ref. 19) but have not pursued their review of his later work on stranded cable modelling.

See:

Roshan Fekr, M., McClure, G., and Farzaneh, M. 1999. Application of ADINA to stress analysis of an optical ground wire. *Computers and Structures*, 72: 301-316.

PhD thesis by Masoud Roshan Fekr: Stress analysis of an optical ground wire. 1999:

[https://mcgill.on.worldcat.org/detailed-](https://mcgill.on.worldcat.org/detailed-record/898040899?databaseList=283&databaseList=638&scope=wz:12129)

[record/898040899?databaseList=283&databaseList=638&scope=wz:12129](https://mcgill.on.worldcat.org/detailed-record/898040899?databaseList=283&databaseList=638&scope=wz:12129)

Gang Qi's modelling work is more recent and also more advanced. It has not been published widely (only in conferences):

Qi, G., and McClure, G. 2013. Advanced computational stress analysis of a stranded overhead line conductor under fretting fatigue conditions. *Proc. 3rd Specialty Conf. On Materials Engineering and Applied Mechanics, CSCE Annual Conference, Montréal, Canada, 29 May - 1 June. Paper MEC-067.*

Qi, G., McClure, G., and Roshan Fekr, M. 2011. Stress Analysis of an Optical Ground Wire using Finite Elements. Abstract (P. 115) and oral presentation at the Sixth M.I.T. Conference on Computational Fluid and Solid Mechanics, Cambridge, MA, USA, 15-17 June.

PhD thesis by Gang Qi: Computational modeling for stress analysis of overhead transmission line stranded conductors under design and fretting fatigue conditions:

[https://mcgill.on.worldcat.org/detailed-](https://mcgill.on.worldcat.org/detailed-record/880373447?databaseList=283&databaseList=638&scope=wz:12129)

[record/880373447?databaseList=283&databaseList=638&scope=wz:12129](https://mcgill.on.worldcat.org/detailed-record/880373447?databaseList=283&databaseList=638&scope=wz:12129)

It is recommended that the authors review these works and indicate how their own studies differ and or agree. At the very least there should be proper citations.

Author's Response to Decision Letter for (RSOS-200309.R0)

See Appendix B.

Decision letter (RSOS-200309.R1)

07-Apr-2020

Dear Dr Liu,

It is a pleasure to accept your manuscript entitled "Study on mechanical characteristics of conductors with 3D finite element models" in its current form for publication in Royal Society Open Science.

on behalf of Dr Derek Abbott (Associate Editor) and R. Kerry Rowe (Subject Editor)
openscience@royalsociety.org

Appendix A

In this paper, refined 3D finite element (FE) models of typical aluminium conductor steel reinforced (ACSR) and formed aluminium conductor steel reinforced (FACSR) with structural details to simulate their static and dynamic characteristics are proposed. The obtained results may provide instructions and foundations for the strength design and type selection of conductors in transmission lines. Some other comments are the following:

- (1) In Fig.9, whether there is slippage between aluminium wires?
- (2) Does the length of the solid conductor models affect the maximum contact pressure of conductors after ice-shedding?
- (3) In section 4. Dynamic characteristics of conductors, why did the solid element choose this area? How are solid elements and beam elements constrained?

- (4) In Fig14, why the amplitude of Mixed model tension increases after 9 seconds? Is there a significant difference in frequency between the Mixed model and Beam model?

(5) In 4.3. Fatigue and wear of conductors, this research result has engineering value. I think, fatigue failure and wear may take place under galloping or wind loads.

Appendix B

Response to Referees:

The authors thank the reviewers very much for their good suggestions, and have tried to respond to the comments and revise the manuscript carefully. The responses to the comments are listed as the follows one by one.

Responses to comments of reviewer 1

In this paper, refined 3D finite element (FE) models of typical aluminium conductor steel reinforced (ACSR) and formed aluminium conductor steel reinforced (FACSR) with structural details to simulate their static and dynamic characteristics are proposed. The obtained results may provide instructions and foundations for the strength design and type selection of conductors in transmission lines. Some other comments are the following:

(1) In Fig.9, whether there is slippage between aluminium wires?

Response: Thank the reviewer very much for his/her comments. Yes, in the 3D finite element models of the conductors, contact condition between the wires is set, so the aluminium wires slip to each other when the conductors are under stretching. A sentence is added to the beginning of Section 3.3.4.

(2) Does the length of the solid conductor models affect the maximum contact pressure of conductors after ice-shedding?

Response: The length of the solid conductor model has little effect on the maximum contact pressure of conductors after ice-shedding. In this paper, the maximum contact pressure is analysed only for the solid elements model part that is enough far away from the connection between the beam conductor model and the solid conductor model. And the contact pressure distribution of this part is less affected by the connection position between the two models. A paragraph is added to the end of Section 4.2.

(3) In section 4. Dynamic characteristics of conductors, why did the solid element choose this area? How are solid elements and beam elements constrained?

Response: (a) According to the numerical results by the FE model discretized by beam elements, it is known that the maximum stress occurs at the vicinity of the suspensions (ref.24), so the solid element model is placed near the suspension. This is explained in the second paragraph of Section 4.1. (b) In the mixed model, each end surface of the solid element model is coupled with a reference point which is connected with the beam node using the kinematic coupling type. Thus, the beam node and the coupling point of solid model have the same displacements. Two sentences are inserted into the second paragraph of Section 4.1.

(4) In Fig14, why the amplitude of Mixed model tension increases after 9 seconds? Is there a significant difference in frequency between the Mixed model and Beam model?

Response: (a) In fact, the dynamic responses of the two models decrease with time totally. The increase of tension amplitude of the Mixed model after 9 second is just disturbing and it decreases to zero finally as shown in the following figure.

Figure. Tension time histories at the middle point of span 2 after ice-shedding.

(b) The frequencies of the mixed model and beam model are nearly the same. The sentence “In the meantime, it is known that the frequencies of the mixed model and the beam model are nearly the same and the relative difference is about 3%, through spectral analysis for the tension time histories curves.” is inserted into the last paragraph of Section 4.1.

(5) In 4.3. Fatigue and wear of conductors, this research result has engineering value. I think, fatigue failure and wear may take place under galloping or wind loads.

Response: The fundamental cause of conductor fatigue failures is the cyclic bending stress. Fatigue failure and wear of conductors often occur due to the wind induced vibration and ice shedding vibration or other causes of vibrations. The evaluation of fatigue lives of conductors in working condition can provide references for the transmission line engineering. A sentence is inserted into paragraph 7 of Section 4.3.

Responses to comments of reviewer 2

(1) The authors present the results of detailed finite element analysis studies of stranded conductors in both static and dynamic regimes of loading. They also compare the response of ACSR conductors and FACSRS conductors and their results indicate that fatigue life of the latter is longer: considering that fatigue life can be tested experimentally (quite routinely by electric utility and cable manufacturers) in a reliable

fashion it is questionable whether high fidelity numerical models are needed for this purpose.

Response: Thank the reviewer very much for his/her careful and constructive suggestions. In this paper, it is important to use a detailed numerical models to analyse the fatigue lives of conductors in the working condition. (a) The electric utility and cable manufacturers usually only perform standard vibration and fatigue tests to verify whether the conductors meet the factory standard. (b) Ice shedding and wind loads on overhead line conductors are usually complicated, and it is difficult to be simulated in laboratory. (c) On the other hand, the fatigue tests are very time-consuming and costly. A new paragraph explaining the idea is inserted after the first paragraph of Section 4.3.

(2) Given the many uncertainties and simplifications used in the ice shedding studies, one would also question the need for more refined cable models to predict the response in terms of cable jumps and wire tensions. Several recent publications on ice shedding from overhead line conductors have addressed such assumptions and important parameters. It does not mean that detailed modelling is wrong, it perhaps is simply not necessary to give additional accuracy to a problem already posed with very inaccurate loading conditions.

Response: Although the loading conditions are inaccurate in the analysed model, the stress distributions estimated by the beam elements, which may be much lower than those by the 3D refined models, are used for the safety design. So it is valuable to carry out the detailed modeling. A sentence is inserted into the first paragraph of Section 4.

(3) In section 1, the authors have ignored two very relevant bodies of work done by Masoud Roshan Fekr et al. and Gang Qi and McClure at McGill University. It seems that the authors have reviewed Roshan Fekr's work on ice shedding (ref. 19) but have not pursued their review of his later work on stranded cable modelling.

See:

Roshan Fekr, M., McClure, G., and Farzaneh, M. 1999. Application of ADINA to stress analysis of an optical ground wire. *Computers and Structures*, 72: 301-316.

PhD thesis by Masoud Roshan Fekr: Stress analysis of an optical ground wire. 1999:

<https://mcgill.on.worldcat.org/detailed->

[record/898040899?databaseList=283&databaseList=638&scope=wz:12129](https://mcgill.on.worldcat.org/detailed-record/898040899?databaseList=283&databaseList=638&scope=wz:12129)

Gang Qi's modelling work is more recent and also more advanced. It has not been published widely (only in conferences):

Qi, G., and McClure, G. 2013. Advanced computational stress analysis of a stranded overhead line conductor under fretting fatigue conditions. Proc. 3rd Specialty Conf. On Materials Engineering and Applied Mechanic, CSCE Annual Conference, Montréal, Canada, 29 May – 1 June. Paper MEC-067.

Qi, G., McClure, G., and Roshan Fekr, M. 2011. Stress Analysis of an Optical Ground Wire using Finite Elements. Abstract (P. 115) and oral presentation at the Sixth M.I.T. Conference on Computational Fluid and Solid Mechanics, Cambridge, MA, USA, 15-17 June.

PhD thesis by Gang Qi: Computational modeling for stress analysis of overhead transmission line stranded conductors under design and fretting fatigue conditions:

<https://mcgill.on.worldcat.org/detailed->

[record/880373447?databaseList=283&databaseList=638&scope=wz:12129](https://mcgill.on.worldcat.org/detailed-record/880373447?databaseList=283&databaseList=638&scope=wz:12129)

It is recommended that the authors review these works and indicate how their own studies differ and or agree. At the very least there should be proper citations.

Response: Thanks for the relevant articles the reviewer recommended. The relevant articles are cited and the difference and agreement from our own are sorted out in this paper. Please see the marked sentences in paragraphs 4 and 5 of Section 1.